# A General Formulation of Independent Policy Optimization in Fully Decentralized MARL

## Abstract

Independent learning is a straightforward solution for fully decentralized learning in cooperative multi-agent reinforcement learning (MARL). The study of independent learning has a history of decades, and the representatives, such as independent Q-learning and independent PPO, can obtain good performance in some benchmarks. However, most independent learning algorithms are without convergence guarantees or theoretical support. In this paper, we propose a general formulation of independent policy optimization, $f$-divergence policy optimization. We show the generality of such a formulation and discuss the limitation. Based on this formulation, we further propose a novel independent learning algorithm, TVPO, that theoretically guarantees convergence. Empirically, we show that TVPO outperforms state-of-the-art fully decentralized learning methods in three popular cooperative MARL benchmarks, which verifies the efficacy of TVPO.

## 1 Introduction

Cooperative multi-agent reinforcement learning (MARL) has shown great potential in many areas including power control (Zhang & Liang, 2020), autonomous vehicle (Han et al., 2022), and robot control (Sartoretti et al., 2019). The mainstream framework for cooperative MARL is centralized training with decentralized Execution (CTDE) (Kraemer & Banerjee, 2016), and the MARL community pays less attention to fully decentralized learning, also known as decentralized training with decentralized execution (DTDE). Fully decentralized learning is still significant in cooperative MARL due to its simplicity. From the perspective of applications, fully decentralized learning is useful in many industrial applications where agents may belong to different parties, e.g., autonomous vehicles or robots. From the perspective of theory, fully decentralized algorithms rely on less information during training and hence are more general and worth further study.

For DTDE or fully decentralized settings, independent learning is a straightforward but effective way, which enables agents to directly execute a single-agent RL algorithm. The representatives are independent Q-learning (IQL) (Tan, 1993) and independent actor-critic (IAC) (Foerster et al., 2018; Papoudakis et al., 2021). Recently, independent PPO (IPPO) (de Witt et al., 2020) extends PPO (Schulman et al., 2017) to MARL and shows good performance in several benchmarks. However, these independent learning algorithms are still troubled by the non-stationarity problem and without convergence guarantees or theoretical support.

In this paper, we propose a general formulation of independent policy optimization, **$f$-divergence policy optimization**. We show the generality of such a formulation for independent learning in cooperative MARL. We also analyze the policy iteration of this formulation and discuss its limitation by a two-player matrix game. Based on this formulation, we further propose a novel independent learning algorithm, **total variation policy optimization** (**TVPO**). To theoretically study the property of TVPO and prove its convergence, we introduce a new set of value functions and policy iteration specifically for fully decentralized learning and prove the monotonicity of this policy iteration. The practical algorithm of TVPO can be effectively realized by an adaptive coefficient, similar to PPO (Schulman et al., 2017).

Empirically, we verify our discussion about the limitation of $f$-divergence policy optimization in the two-player matrix game and show the joint policy may converge to the sub-optimum with different $f$-divergences. Moreover, we evaluate the performance of TVPO in three popular benchmarks of

cooperative MARL including MPE (Lowe et al., 2017), SMAC (Samvelyan et al., 2019), and multi-agent MuJoCo (Peng et al., 2021). We compare TVPO with four representative fully decentralized learning methods: IQL (Tan, 1993), IPPO (de Witt et al., 2020), I2Q (Jiang & Lu, 2022), and DPO (Su & Lu, 2022b). The empirical results show that TVPO outperforms these baselines in all evaluated tasks, which verifies the effectiveness of TVPO in fully decentralized cooperative MARL.

## 2  RELATED WORK

**CTDE.** Centralized training with decentralized execution (CTDE) is the mainstream framework for solving cooperative MARL problems (Lowe et al., 2017; Foerster et al., 2018; Sunehag et al., 2018; Rashid et al., 2018; Iqbal & Sha, 2019; Wang et al., 2021a; Zhang et al., 2021; Su & Lu, 2022a; Wang et al., 2023a). CTDE settles the non-stationarity problem through centralized training. This line of research can be divided into two kinds: one is value decomposition algorithms (Sunehag et al., 2018; Rashid et al., 2018; Son et al., 2019; Yang et al., 2020; Wang et al., 2021a), where the optimum of the centralized Q-function corresponds to the optimum of the decentralized Q-functions so the learning of the centralized Q-function can be factorized into the learning of the decentralized Q-functions; the other is multi-agent actor-critic algorithms (Foerster et al., 2018; Iqbal & Sha, 2019; Wang et al., 2021b; Zhang et al., 2021; Su & Lu, 2022a; Wang et al., 2023a), in which a centralized Q-function is used for the learning of decentralized policies. TRPO (Schulman et al., 2015) and PPO (Schulman et al., 2017) are also extended into the MARL setting by HAPPO (Kuba et al., 2021) and MAPPO (Yu et al., 2021) respectively via learning a centralized state value function. ***However, all these methods are CTDE and thus inappropriate for fully decentralized learning.***

**Fully Decentralized Learning.** There are several different views recently about fully decentralized learning or decentralized learning. Some works study decentralized learning specifically with communication (Zhang et al., 2018; Li et al., 2020) or parameter sharing (Terry et al., 2020). Actually, both communication and parameter sharing exchange information among agents (Terry et al., 2020). ***However, in this paper, we consider fully decentralized learning in the strictest sense – with each agent independently learning its policy while being not allowed to communicate or share parameters*** as in Tampuu et al. (2015); Mao et al. (2022); Wang et al. (2023b). Independent learning (OroojlooyJadid & Hajinezhad, 2019) is the most straightforward approach for fully decentralized learning and has actually been a subject of study in cooperative MARL for decades. The representatives are independent Q-learning (IQL) (Tan, 1993; Tampuu et al., 2015), independent actor-critic (IAC) as Foerster et al. (2018); Papoudakis et al. (2021), and independent PPO (IPPO) (de Witt et al., 2020). All these independent learning algorithms violate the stationary condition of MDP and do not have convergence guarantees, though IQL and IPPO obtain good performance in several benchmarks (Papoudakis et al., 2021). There are two recent studies, I2Q (Jiang & Lu, 2022) and DPO (Su & Lu, 2022b), with convergence guarantees in fully decentralized MARL. I2Q introduces QSS-value (Edwards et al., 2020) into independent Q-learning and obtains the convergence guarantee, but is limited to deterministic environments. DPO proposes a decentralized surrogate of the joint TRPO objective to obtain the convergence guarantee. ***Empirically, I2Q shows better performance than IQL, while DPO outperforms IPPO. We will compare TVPO with these state-of-the-art methods in our empirical studies.***

## 3  PRELIMINARIES

**Dec-POMDP.** The decentralized partially observable Markov decision process (Dec-POMDP) is a general model for cooperative MARL. A Dec-POMDP is defined as a tuple $\mathcal{G} = \{S, A, P, Y, O, I, N, r, \gamma\}$. $N$ is the number of agents, and $I = \{1, 2 \cdots N\}$ is the set of all agents. $S$ is the state space. $A = A_1 \times A_2 \times \cdots \times A_N$ represents the joint action space, where $A_i$ is the individual action space for agent $i$. $P(s'|s, \boldsymbol{a}) : S \times A \times S \rightarrow [0, 1]$ is the transition function. $Y$ is the observation space, and $O(s, i) : S \times I \rightarrow Y$ is a mapping from state to observation for each agent $i$. $\gamma \in [0, 1)$ is the discount factor, and $r(s, \boldsymbol{a}) : S \times A \rightarrow [-r_{\max}, r_{\max}]$ is the reward function of state $s \in S$ and joint action $\boldsymbol{a} \in A$, where $r_{\max}$ is the bound of the reward function. The objective of Dec-POMDP is to maximize $J(\boldsymbol{\pi}) = \mathbb{E}_{\boldsymbol{\pi}} \left[ \sum_{t=0} \gamma^t r(s_t, \boldsymbol{a}_t) \right]$, thus we need to find the optimal joint policy $\boldsymbol{\pi}^* = \arg \max_{\boldsymbol{\pi}} J(\boldsymbol{\pi})$. To settle the partial observable problem, history $\tau_i \in \mathcal{T}_i = (Y \times A_i)^*$ is often used to replace observation $o_i \in Y$. In fully decentralized learning, each agent $i$ independently learns an individual policy $\pi^i(a_i|\tau_i)$ and their joint policy $\boldsymbol{\pi}$ can be

represented as the product of each $\pi^i$. ***Though each agent learns individual policy as $\pi^i(a_i|\tau_i)$ in practice, in our analysis, we assume that each agent receives the state $s$, following existing studies*** (Jiang & Lu, 2022; Su & Lu, 2022b), ***because the analysis in partially observable environments is much hard and the problem may be undecidable in Dec-POMDP*** (Madani et al., 1999). Moreover, the V-function and Q-function of the joint policy $\boldsymbol{\pi}$ are as follows,

$$V^{\boldsymbol{\pi}}(s) = \mathbb{E}_{\boldsymbol{a} \sim \boldsymbol{\pi}}\left[Q^{\boldsymbol{\pi}}(s, \boldsymbol{a})\right] \tag{1}$$

$$Q^{\boldsymbol{\pi}}(s, \boldsymbol{a}) = r(s, \boldsymbol{a}) + \gamma \mathbb{E}_{s' \sim P(\cdot|s, \boldsymbol{a})}\left[V^{\boldsymbol{\pi}}(s')\right]. \tag{2}$$

**Fully Decentralized Critic.** The critic in fully decentralized learning or independent learning has been discussed in previous studies such as Peshkin et al. (2000); Lyu & Xiao (2021). However, for the convenience of further discussion, we provide some formulations and deductions about the fully decentralized critic.

In fully decentralized learning, each agent learns independently from its own interactions with the environment. Therefore, the Q-function of each agent $i$ is actually the following formula:

$$Q_{\pi^{-i}}^{\pi^i}(s, a_i) = r_{\pi^{-i}}(s, a_i) + \gamma \mathbb{E}_{a_{-i} \sim \pi^{-i}, s' \sim P(\cdot|s, a_i, a_{-i}), a_i' \sim \pi^i}[Q_{\pi^{-i}}^{\pi^i}(s', a_i')], \tag{3}$$

where $r_{\pi^{-i}}(s, a_i) = \mathbb{E}_{\pi^{-i}}[r(s, a_i, a_{-i})]$, and $\pi^{-i}$ and $a_{-i}$ respectively denote the joint policy and joint action of all agents expect agent $i$. If we take the expectation $\mathbb{E}_{a'_{-i} \sim \pi^{-i}(\cdot|s'), a_{-i} \sim \pi^{-i}(\cdot|s)}$ over both sides of the Q-function of the joint policy (2), then we have

$$\mathbb{E}_{\pi^{-i}}[Q^{\boldsymbol{\pi}}(s, a_i, a_{-i})] = r_{\pi^{-i}}(s, a_i) + \gamma \mathbb{E}_{a_{-i} \sim \pi^{-i}, s' \sim P(\cdot|s, a_i, a_{-i}), a_i' \sim \pi^i}\left[\mathbb{E}_{\pi^{-i}}[Q^{\boldsymbol{\pi}}(s', a_i', a_{-i}')]\right].$$

We can see that $Q_{\pi^{-i}}^{\pi^i}(s, a_i)$ and $\mathbb{E}_{\pi^{-i}}[Q^{\boldsymbol{\pi}}(s, a_i, a_{-i})]$ satisfy the same iteration. Moreover, we show in the following that $Q_{\pi^{-i}}^{\pi^i}(s, a_i)$ and $\mathbb{E}_{\pi^{-i}}[Q^{\boldsymbol{\pi}}(s, a_i, a_{-i})]$ are just the same.

We first define an operator $\Gamma_{\pi^{-i}}^{\pi^i}$ as follows,

$$\Gamma_{\pi^{-i}}^{\pi^i} Q(s, a_i) = r_{\pi^{-i}}(s, a_i) + \gamma \mathbb{E}_{a_{-i} \sim \pi^{-i}, s' \sim P(\cdot|s, a_i, a_{-i}), a_i' \sim \pi^i}[Q(s', a_i')].$$

Then we prove that the operator $\Gamma_{\pi^{-i}}^{\pi^i}$ is a contraction. Considering any two individual Q-functions $Q_1$ and $Q_2$, we have:

$$\|\Gamma_{\pi^{-i}}^{\pi^i} Q_1 - \Gamma_{\pi^{-i}}^{\pi^i} Q_2\|_\infty = \max_{s, a_i} \gamma |\mathbb{E}_{a_{-i}, s', a_i'}[Q_1(s', a_i') - Q_2(s', a_i')]|$$

$$\leq \gamma \mathbb{E}_{a_{-i}, s', a_i'}[\max_{s', a_i'} |Q_1(s', a_i') - Q_2(s', a_i')|] = \gamma \max_{s', a_i'} |Q_1(s', a_i') - Q_2(s', a_i')|$$

$$= \gamma \|Q_1 - Q_2\|_\infty.$$

So the operator $\Gamma_{\pi^{-i}}^{\pi_i}$ has one and only one fixed point, which means

$$Q_{\pi^{-i}}^{\pi^i}(s, a_i) = \mathbb{E}_{\pi^{-i}}[Q^{\boldsymbol{\pi}}(s, a_i, a_{-i})], \quad V_{\pi^{-i}}^{\pi^i}(s) = \mathbb{E}_{\pi^{-i}}[V^{\boldsymbol{\pi}}(s)] = V^{\boldsymbol{\pi}}(s),$$

and the fully decentralized critic (3) is well-defined. For simplicity, in the following, we use $Q_i^{\boldsymbol{\pi}}$ to denote $Q_{\pi^{-i}}^{\pi^i}$ given a joint policy $\boldsymbol{\pi}$, if there is no confusion.

**Independent Learning.** Independent learning is a straightforward method to solve cooperative MARL problems, which makes each agent learn through the same single-agent RL algorithm, such IQL (Tan, 1993), IAC (Foerster et al., 2018), and IPPO (de Witt et al., 2020). Though independent learning faces the non-stationarity problem, it still has the advantage of absorbing the benefit of single-agent RL. Policy iteration $\pi_{\text{new}} = \arg\max_\pi \sum_a \pi(a|s)Q^{\pi_{\text{old}}}(s, a)$ is fundamental in single-agent RL, which ensures that $\pi_{\text{new}}$ improves monotonically over $\pi_{\text{old}}$ and guarantees the convergence. We draw inspiration from policy iteration in single-agent RL, introduce a general formulation of independent policy optimization, and try to find an independent learning algorithm that can guarantee convergence in cooperative MARL.

## 4 A GENERAL FORMULATION FOR INDEPENDENT POLICY OPTIMIZATION

Given the condition of fully decentralized learning in cooperative MARL, we first propose a general formulation of independent policy optimization, $f$-divergence policy optimization, and discuss its generality and limitation. Then, based on this formulation, we propose total variation policy optimization (TVPO), prove the convergence of TVPO in fully decentralized learning, and provide the practical algorithm.

Before diving into the discussion, we need to introduce a simple two-player matrix game which will be used later. In this matrix game, the two agents, Alice and Bob, both have two actions and we denote them as $\{u_A^0, u_A^1\}$ for Alice and $\{u_B^0, u_B^1\}$ for Bob. Each episode of this matrix game has only one step. The rewards for the joint actions $(u_A^0, u_B^0)$, $(u_A^0, u_B^1)$, $(u_A^1, u_B^0)$ and $(u_A^1, u_B^1)$ are $a$, $b$, $c$, and $d$ respectively. The policies of Alice and Bob can be described with $p_t$ and $q_t$ as that Alice will take action $u_A^0$ with probability $p_t$ and Bob will take action $u_B^0$ with probability $q_t$, where $t$ represents the number of policy iterations. The full information of this matrix game is illustrated in Table 1.

|  |  | $u_B^0$ | $u_B^1$ |
|---|---|---|---|
| Bob / Alice |  | $q_t$ | $1-q_t$ |
| $u_A^0$ | $p_t$ | $a$ | $b$ |
| $u_A^1$ | $1-p_t$ | $c$ | $d$ |

Table 1: The two-player matrix game for Alice and Bob with policies after the number $t$ of policy iterations. Alice will take action $u_A^0$ with probability $p_t$ and take action $u_A^1$ with probability $1 - p_t$; Bob will take action $u_B^0$ with probability $q_t$ and take action $u_B^1$ with probability $1 - q_t$.

## 4.1 $f$-DIVERGENCE POLICY OPTIMIZATION

The $f$-divergence policy optimization is formulated as follows,

$$\pi_{\text{new}}^i = \arg\max_{\pi^i} \sum_{a_i} \pi^i(a_i|s) Q_i^{\pi_{\text{old}}}(s, a_i) - \omega D_f\left(\pi^i(\cdot|s) \| \pi_{\text{old}}^i(\cdot|s)\right) \tag{4}$$

$$= \arg\max_{\pi^i} \sum_{a_i} \pi^i(a_i|s) Q_i^{\pi_{\text{old}}}(s, a_i) - \omega \sum_{a_i} \pi_{\text{old}}^i(a_i|s) f\left(\frac{\pi^i(a_i|s)}{\pi_{\text{old}}^i(a_i|s)}\right), \tag{5}$$

where $D_f(p\|q) \triangleq \sum_i q_i f\left(\frac{p_i}{q_i}\right)$ is $f$-divergence and according to the definition of $f$-divergence, $f : [0, \infty) \to (-\infty, +\infty]$ is convex and $f(1) = 0$. This formulation contains an additional term $D_f\left(\pi^i(\cdot|s) \| \pi_{\text{old}}^i(\cdot|s)\right)$, which describes the distance between $\pi^i$ and $\pi_{\text{old}}^i$.

There has been several studies considering the distance between $\boldsymbol{\pi}_{\text{old}}$ and $\boldsymbol{\pi}_{\text{new}}$. The trust region in TRPO (Schulman et al., 2015) and PPO (Schulman et al., 2017) is actually KL-divergence between $\boldsymbol{\pi}_{\text{old}}$ and $\boldsymbol{\pi}_{\text{new}}$, while Nachum et al. (2017) extend entropy regularization to a more general formulation with KL-divergence. Unlike these studies that just use KL-divergence as the distance measure, we would like to discuss a more general formulation. So we use $f$-divergence, which is widely used for describing the distance between two distributions. Also, KL-divergence is a special case of $f$-divergence with $f(x) = x \log x$ and we have many other choices for $f$-divergence such as $f(x) = \frac{|x-1|}{2}$ corresponding to total variation distance $D_f(p\|q) = \frac{1}{2}\sum_i |p_i - q_i|$ and $f(x) = (1 - \sqrt{x})^2$ corresponding to Hellinger distance $D_f(p\|q) = \sqrt{\sum_i(\sqrt{p_i} - \sqrt{q_i})^2}$.

To further discuss $f$-divergence policy optimization, we need to find the solution to the optimization objective (4) and we have the following lemma.

**Lemma 1.** *Given a fixed function $f$ and the corresponding $f$-divergence $D_f$, let $g(x) = (f')^{-1}(x)$, then the solution to (4) is*

$$\pi_{\text{new}}^i(a_i|s) = \max\{\pi_{\text{old}}^i(a_i|s) g\left(\frac{\lambda_s + Q_i^{\pi_{\text{old}}}(s, a_i)}{\omega}\right), 0\}, \tag{6}$$

*where $\lambda_s$ satisfies $\sum_{a_i} \max\{\pi_{\text{old}}^i(a_i|s) g\left(\frac{\lambda_s + Q_i^{\pi_{\text{old}}}(s, a_i)}{\omega}\right), 0\} = 1$.*

This proof is included in Appendix A.1 and follows Yang et al. (2019).

We use the two-player matrix game between Alice and Bob (i.e., Table 1) to discuss the limitation of $f$-divergence policy optimization. As for the policy iteration in the matrix game, we have the following proposition.

**Proposition 1.** *Suppose $g(x) \geq 0$ and let $M = b + c - a - d$, $\hat{p} = \frac{c-d}{M}$, and $\hat{q} = \frac{b-d}{M}$. If the payoff matrix of the two-player matrix game satisfies $M > 0$, and Alice and Bob update their policies with*

$$\pi_{t+1}^i = \arg\max_{\pi^i} \sum_{a_i} \pi^i(a_i|s) Q_i^{\pi_t}(s, a_i) - \omega D_f\left(\pi^i(\cdot|s) \| \pi_t^i(\cdot|s)\right), \tag{7}$$

*then we have (1) $p_t > \hat{p} \Rightarrow q_{t+1} < q_t$; (2) $p_t < \hat{p} \Rightarrow q_{t+1} > q_t$; (3) $q_t > \hat{q} \Rightarrow p_{t+1} < p_t$; (4) $q_t < \hat{q} \Rightarrow p_{t+1} > p_t$.*

The proof is included in Appendix A.2. With Proposition 1, we can build a case where the joint policy sequence can only converge to the sub-optimum. We assume the matrix game satisfies the condition $b > c > \max\{a, d\}$, then the optimal joint policy is $(p_t, q_t) = (1, 0)$ corresponding to the joint action $(u_A^0, u_B^1)$ and reward $b$. Moreover, the condition $b > c > \max\{a, d\}$ also means $\hat{p} \in (0, 1)$ and $\hat{q} \in (0, 1)$. If at iteration $t$, the condition $q_t > \hat{q}$, $p_t < \hat{p}$ is satisfied, then $q_{t+1} > q_t > \hat{q}$, $p_{t+1} < p_t < \hat{p}$. By induction, we know that $\forall t' \geq t$, $q_{t'+1} > q_{t'} > \hat{q}, p_{t'+1} < p_{t'} < \hat{p}$. As the sequence $\{p_t\}$ and $\{q_t\}$ are both bounded in the interval $[0, 1]$, we know the sequence $\{p_t\}$ and $\{q_t\}$ will converge to $p^*$ and $q^*$. As for $p^*$ and $q^*$, we have the following corollary.

**Corollary 1.** *If at iteration $t$, the condition $q_t > \hat{q}$, $p_t < \hat{p}$ is satisfied, then the sequence $\{p_t\}$ and $\{q_t\}$ will converge to $p^* = 0$ and $q^* = 1$ respectively.*

The proof is included in Appendix A.3. Corollary 1 tells us if once $q_t > \hat{q}$, $p_t < \hat{p}$, then the joint policy will converge to the sub-optimal solution $(p^*, q^*) = (0, 1)$ corresponding to the joint action $(u_A^1, u_B^0)$ and reward $c$. So if the initial policy $p_0$ and $q_0$ satisfies the condition $q_0 > \hat{q}$, $p_0 < \hat{p}$, then the joint policy will converge to the sub-optimal policy. We will further illustrate this in the experiment.

### 4.2 TOTAL VARIATION POLICY OPTIMIZATION

The $f$-divergence formulation (4) is trapped in the sub-optimal joint policy even in a simple two-player matrix game. This shows the upper bound of $f$-divergence policy optimization, so we should not expect such a policy iteration could obtain the optimal joint policy in fully decentralized learning. Fortunately, we have found an algorithm that accords with the $f$-divergence formulation and has the convergence guarantee. This algorithm uses total variation distance for $f$-divergence, so we call it total variation policy optimization (TVPO). The convergence guarantee of TVPO shows the potential of the $f$-divergence formulation.

Before we introduce TVPO and prove its convergence, we need some definitions and lemma. We use $D_{\text{TV}}(p \| q) \triangleq \frac{1}{2} \sum_i |p_i - q_i|$ to represent total variation distance. We define a new V-function $V_{\boldsymbol{\rho}}^{\boldsymbol{\pi}}(s)$ and a new Q-function $Q_{\boldsymbol{\rho}}^{\boldsymbol{\pi}}(s, a_i, a_{-i})$ given joint polices $\boldsymbol{\pi}$ and $\boldsymbol{\rho}$ as follows:

$$V_{\boldsymbol{\rho}}^{\boldsymbol{\pi}}(s) = \frac{1}{N} \sum_i \sum_{a_i} \pi^i(a_i|s) \sum_{a_{-i}} \rho^{-i}(a_{-i}|s) Q_{\boldsymbol{\rho}}^{\boldsymbol{\pi}}(s, a_i, a_{-i}) - \omega D_f\left(\pi^i(\cdot|s) \| \rho^i(\cdot|s)\right), \quad (8)$$

$$Q_{\boldsymbol{\rho}}^{\boldsymbol{\pi}}(s, a_i, a_{-i}) = r(s, a_i, a_{-i}) + \gamma \mathbb{E}_{s' \sim P(\cdot|s, a_i, a_{-i})}\left[V_{\boldsymbol{\rho}}^{\boldsymbol{\pi}}(s')\right]. \quad (9)$$

As the definition (8) is a fixed-point equation, we need to prove that this definition is well-defined. So we define an operator $\Gamma_{\boldsymbol{\rho}}^{\boldsymbol{\pi}}$ as follows:

$$\Gamma_{\boldsymbol{\rho}}^{\boldsymbol{\pi}} V(s) = \frac{1}{N} \sum_i \sum_{a_i} \pi^i(a_i|s) \sum_{a_{-i}} \rho^{-i}(a_{-i}|s) \left(r(s, a_i, a_{-i}) + \gamma \mathbb{E}\left[V(s')\right]\right) - \omega D_f\left(\pi^i(\cdot|s) \| \rho^i(\cdot|s)\right).$$

Then for any value function $V_1$ and $V_2$, we have

$$\left\|\Gamma_{\boldsymbol{\rho}}^{\boldsymbol{\pi}} V_1(s) - \Gamma_{\boldsymbol{\rho}}^{\boldsymbol{\pi}} V_2(s)\right\|_{\infty} = \left\|\frac{1}{N} \sum_i \sum_{a_i} \pi^i(a_i|s) \sum_{a_{-i}} \rho^{-i}(a_{-i}|s) \left(\gamma \mathbb{E}\left[V_1(s')\right] - \gamma \mathbb{E}\left[V_2(s')\right]\right)\right\|_{\infty}$$

$$\leq \gamma \|V_1(s) - V_2(s)\|_{\infty}.$$

So the operator $\Gamma_{\boldsymbol{\rho}}^{\boldsymbol{\pi}}$ is a $\gamma$-contraction, which means $V_{\boldsymbol{\rho}}^{\boldsymbol{\pi}}(s)$ is the unique fixed-point of (8) and the definition (8) is well-defined.

To apply total variation distance to independent policy optimization, we have the following lemma.

**Lemma 2.** *Suppose $\boldsymbol{\pi}_{\text{new}}$, $\boldsymbol{\pi}_{\text{old}}$, and $\boldsymbol{\pi}$ are three joint policies. Let $M = \frac{2r_{\max}}{1-\gamma}$, then for any state $s$, we have*

$$\sum_{\boldsymbol{a}} \boldsymbol{\pi}_{\text{new}}(\boldsymbol{a}|s) Q^{\boldsymbol{\pi}}(s, \boldsymbol{a}) \geq \frac{1}{N} \sum_{i=1}^N \sum_{a_i} \pi_{\text{new}}^i(a_i|s) \sum_{a_{-i}} \pi_{\text{old}}^{-i}(a_{-i}|s) Q^{\boldsymbol{\pi}}(s, a_i, a_{-i})$$

$$- \frac{(N-1)M}{N} \sum_{i=1}^N D_{\text{TV}}\left(\pi_{\text{new}}^i(\cdot|s) \| \pi_{\text{old}}^i(\cdot|s)\right). \quad (10)$$

The proof is included in Appendix A.4. Lemma 2 is a critical bridge between normal value function $V^{\boldsymbol{\pi}}$ and our new value function $V_{\boldsymbol{\rho}}^{\boldsymbol{\pi}}$, and we can witness its effect in our later discussion. Moreover, we also know that $V_{\boldsymbol{\pi}}^{\boldsymbol{\pi}} = V^{\boldsymbol{\pi}}$ and $Q_{\boldsymbol{\pi}}^{\boldsymbol{\pi}} = Q^{\boldsymbol{\pi}}$.

We can also realize the monotonic improvement with a fully decentralized optimization objective via the following proposition.

**Proposition 2.** *Given a fixed joint policy $\boldsymbol{\rho}$ and an old joint policy $\boldsymbol{\pi}_{\text{old}}$, if all the agents update their policies according to*

$$\pi_{\text{new}}^i = \arg\max_{\pi^i} \sum_{a_i} \pi^i(a_i|s) \sum_{a_{-i}} \rho^{-i}(a_{-i}|s) Q_{\boldsymbol{\rho}}^{\boldsymbol{\pi}_{\text{old}}}(s, a_i, a_{-i}) - \omega D_f \left( \pi^i(\cdot|s) \| \rho^i(\cdot|s) \right), \quad (11)$$

*then we have $V_{\boldsymbol{\rho}}^{\boldsymbol{\pi}_{\text{old}}}(s) \leq V_{\boldsymbol{\rho}}^{\boldsymbol{\pi}_{\text{new}}}(s)$, $Q_{\boldsymbol{\rho}}^{\boldsymbol{\pi}_{\text{old}}}(s, \boldsymbol{a}) \leq Q_{\boldsymbol{\rho}}^{\boldsymbol{\pi}_{\text{new}}}(s, \boldsymbol{a})$ $\forall s \in S, \boldsymbol{a} \in A$.*

The proof is included in Appendix A.5. According to (11), by taking $\boldsymbol{\pi}_{\text{old}} = \boldsymbol{\rho} = \boldsymbol{\pi}_t$ and $\boldsymbol{\pi}_{\text{new}} = \boldsymbol{\pi}_{t+1}$, we can design a policy iteration as follows:

$$\pi_{t+1}^i = \arg\max_{\pi^i} \sum_{a_i} \pi^i(a_i|s) \sum_{a_{-i}} \pi_t^{-i}(a_{-i}|s) Q^{\boldsymbol{\pi}_t}(s, a_i, a_{-i}) - \omega D_f \left( \pi^i(\cdot|s) \| \pi_t^i(\cdot|s) \right). \quad (12)$$

This policy iteration resolves the $f$-divergence formulation (4). According to Proposition 2, we know the joint policy sequence $\{\boldsymbol{\pi}_t\}$ has the property $V_{\boldsymbol{\pi}_t}^{\boldsymbol{\pi}_{t+1}}(s) \geq V_{\boldsymbol{\pi}_t}^{\boldsymbol{\pi}_t}(s) = V^{\boldsymbol{\pi}_t}(s)$. By taking $D_f = D_{\text{TV}}$ and $\omega = \frac{(N-1)M}{N}$, we can combine these results with Lemma 2 to obtain the convergence guarantee.

**Theorem 1.** *Let $\omega = \frac{(N-1)M}{N}$. If all agents update their policies according to*

$$\pi_{t+1}^i = \arg\max_{\pi^i} \sum_{a_i} \pi^i(a_i|s) \sum_{a_{-i}} \pi_t^{-i}(a_{-i}|s) Q^{\boldsymbol{\pi}_t}(s, a_i, a_{-i}) - \omega D_{\text{TV}} \left( \pi^i(\cdot|s) \| \pi_t^i(\cdot|s) \right)$$

$$= \arg\max_{\pi^i} \sum_{a_i} \pi^i(a_i|s) Q_i^{\boldsymbol{\pi}_t}(s, a_i) - \omega D_{\text{TV}} \left( \pi^i(\cdot|s) \| \pi_t^i(\cdot|s) \right), \quad (13)$$

*then we have $V_{\boldsymbol{\pi}_t}^{\boldsymbol{\pi}_{t+1}}(s) \geq V^{\boldsymbol{\pi}_t}(s) \geq V_{\boldsymbol{\pi}_{t-1}}^{\boldsymbol{\pi}_t}(s) \geq V^{\boldsymbol{\pi}_{t-1}}(s)$. Moreover, the sequence $\{V^{\boldsymbol{\pi}_t}\}$ and $\{\boldsymbol{\pi}_t\}$ will converge to $V^*$ and $\boldsymbol{\pi}_*$ respectively, which satisfy the fixed-point equation*

$$\pi_*^i = \arg\max_{\pi^i} \sum_{a_i} \pi^i(a_i|s) \sum_{a_{-i}} \pi_*^{-i}(a_{-i}|s) \left( r(s, a_i, a_{-i}) + \gamma \mathbb{E} \left[ V^*(s') \right] \right) - \omega D_{\text{TV}} \left( \pi^i(\cdot|s) \| \pi_*^i(\cdot|s) \right).$$

The proof is included in Appendix A.6.

**Remark.** The policy optimization objective of TVPO is (13). An important property of (13) is that it can be optimized individually and independently by each agent and the joint policy will converge according to Theorem 1. Although (13) is similar to the surrogate of DPO (Su & Lu, 2022b), there are two main differences between TVPO and DPO. The first difference is that from the property $D_{\text{TV}}^2(p\|q) \leq D_{\text{KL}}(p\|q)$, the bound $D_{\text{TV}}$ of TVPO is tighter than $\sqrt{D_{\text{KL}}}$ in DPO. The second difference is that TVPO obtains the convergence guarantee through policy iteration while DPO obtains the convergence guarantee through the surrogate of joint TRPO objective. We will investigate their empirical performance in the experiment.

### 4.3 THE PRACTICAL ALGORITHM OF TVPO

Practically, if we use the objective (13) directly, then the large coefficient $\omega$ will greatly limit the step size of the policy update, and the algorithm will not work (Schulman et al., 2015). So we follow previous studies such as PPO (Schulman et al., 2017) to use an adaptive coefficient $\beta^i$ to replace $\omega$, then the policy optimization objective can be rewritten as

$$\pi_{t+1}^i = \arg\max_{\pi^i} \sum_{a_i} \pi^i(a_i|s) A_i^{\boldsymbol{\pi}_t}(s, a_i) - \beta^i D_{\text{TV}} \left( \pi^i(\cdot|s) \| \pi_t^i(\cdot|s) \right), \quad (14)$$

where $A_i^{\boldsymbol{\pi}_t}(s, a_i) = Q_i^{\boldsymbol{\pi}_t}(s, a_i) - \mathbb{E}_{\pi_t^i} [Q_i^{\boldsymbol{\pi}_t}(s, a_i)] = Q_i^{\boldsymbol{\pi}_t}(s, a_i) - V^{\boldsymbol{\pi}_t}(s)$. Here we use the baseline $V^{\boldsymbol{\pi}_t}(s)$ to reduce the variance in training.

---

**Algorithm 1.** The practical algorithm of TVPO

---

1: **for** episode = 1 to $M$ **do**
2:     **for** $t = 1$ to max_episode_length **do**
3:         select action $a_i \sim \pi^i(\cdot|s)$
4:         execute $a_i$ and observe reward $r$ and next state $s'$
5:         collect $\langle s, a_i, r, s' \rangle$
6:     **end for**
7:     Update the critic according to (16)
8:     Update the policy according to (14) or (17)
9:     Update $\beta^i$ according to (15).
10: **end for**

---

The update rule of $\beta^i$ follows the practice of PPO. We can choose a hyperparameter $d_{\text{target}}$ which means we expect the total variation distance should be around $d_{\text{target}}$. Then we can update $\beta^i$ according to the value of $D_{\text{TV}}\left(\pi_{t+1}^i(\cdot|s)\|\pi_t^i(\cdot|s)\right)$ in training as follows:

$$
\begin{aligned}
&\text{if } D_{\text{TV}}\left(\pi_{t+1}^i(\cdot|s)\|\pi_t^i(\cdot|s)\right) > d_{target} * \delta, \quad \text{then } \beta^i \leftarrow \beta^i * \alpha \\
&\text{if } D_{\text{TV}}\left(\pi_{t+1}^i(\cdot|s)\|\pi_t^i(\cdot|s)\right) < d_{target}/\delta, \quad \text{then } \beta^i \leftarrow \beta^i/\alpha,
\end{aligned}
\tag{15}
$$

where $\delta$ and $\alpha$ are two constants and we choose $\delta = 1.5$ and $\alpha = 2$ like the choice of PPO.

For the critic, since the policy update needs to calculate $A_i^{\boldsymbol{\pi}_t}(s, a_i) = \mathbb{E}_{\pi_t^{-i}}[r(s, a_i, a_{-i}) + \gamma V^{\boldsymbol{\pi}_t}(s') - V^{\boldsymbol{\pi}_t}(s)]$, we take an individual state value function $V^i(s)$ as the critic for each agent $i$ and approximate $A_i^{\boldsymbol{\pi}_t}(s, a_i)$ with $\hat{A}_i = r + \gamma V^i(s') - V^i(s)$. The critic is updated as follows:

$$
\mathcal{L}_{\text{critic}}^i = \mathbb{E}\left[(V^i(s) - y_i)^2\right], \quad \text{where } y_i = r + \gamma V^i(s') \text{ or other target values.}
\tag{16}
$$

When facing continuous action space, we usually use Gaussian distribution as the policy. However, there is no closed-form solution for total variation distance between two Gaussian distributions, to the best of our knowledge. To avoid optimization difficulties, we replace total variation distance with Hellinger distance $D_{\text{H}}(p\|q) = \sqrt{\sum_i(\sqrt{p_i} - \sqrt{q_i})^2}$ in the environment with continuous action space, since there is a closed-form solution for Hellinger distance between two Gaussian distributions. Moreover, Hellinger distance has a critical property related to total variation distance that $D_{\text{TV}}(p\|q) \le D_{\text{H}}(p\|q)$ and the proof is included in Appendix A.7.

With this property, we can replace $D_{\text{TV}}$ with $D_{\text{H}}$ in Lemma 2 and Theorem 1, while we can still obtain the same convergence guarantee. Thus, for the continuous action space, we use the following policy optimization objective:

$$
\pi_{t+1}^i = \arg\max_{\pi^i} \sum_{a_i} \pi^i(a_i|s) A_i^{\boldsymbol{\pi}_t}(s, a_i) - \beta^i D_{\text{H}}\left(\pi^i(\cdot|s)\|\pi_t^i(\cdot|s)\right).
\tag{17}
$$

The practical algorithm of TVPO is summarized in Algorithm 1.

## 5 EXPERIMENTS

The experiment contains two main parts. The first part is to verify the limitation of $f$-divergence policy optimization as we have discussed in Section 4.1 through the matrix game. The second part is

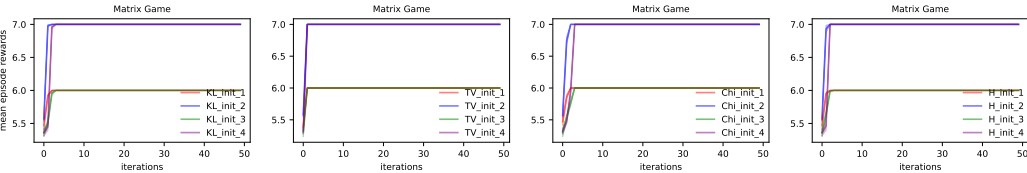

Figure 1: Learning curves of KL-iteration, TV-iteration, $\chi^2$-iteration, and H-iteration over four different sets of initialization in the matrix game.

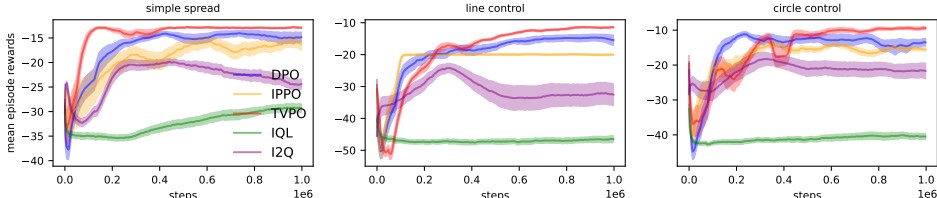

Figure 2: Learning curves of TVPO compared with IQL, IPPO, I2Q, and DPO in 10-agent simple spread, 10-agent line control, and 10-agent circle control in MPE.

to evaluate the performance of TVPO in three popular cooperative MARL benchmarks including MPE (Lowe et al., 2017), SMAC (Samvelyan et al., 2019), and multi-agent MuJoCo (Peng et al., 2021), compared with state-of-the-art fully decentralized algorithms. All learning curves correspond to five different random seeds and the shaded area corresponds to the 95% confidence interval.

## 5.1 Verification in Matrix Game

In this section, we choose $a = 5$, $b = 7$, $c = 6$, $d = 4$ for the matrix game, which satisfies the condition $b > c > \max\{a, d\}$ as mentioned in Section 4.1. We use four different specific $f$-divergences: KL-divergence, total variation distance, $\chi^2$-distance, and Hellinger distance to build four different iterations of (4). We call these four iterations as KL-iteration, TV-iteration, $\chi^2$-iteration, and H-iteration respectively. We test these iterations over four sets of initialization: init_1 $(p_0, q_0) = (0.4, 0.8)$; init_2 $(p_0, q_0) = (0.6, 0.6)$; init_3 $(p_0, q_0) = (0.49, 0.76)$; init_4 $(p_0, q_0) = (0.51, 0.74)$. For the matrix game, we can calculate that $(\hat{p}, \hat{q}) = (0.5, 0.75)$ as defined in Proposition 1. From the discussion in Section 4.1 we know that init_1 and init_3 satisfy the condition $p_0 < \hat{p}$, $q_0 > \hat{q}$, which means the converged policy should be the sub-optimal policy $(p^*, q^*) = (0, 1)$ with reward $c = 6$, and init_2 and init_4 satisfy the condition $p_0 > \hat{p}$, $q_0 < \hat{q}$, which means the converged policy should be the optimal policy $(p^*, q^*) = (1, 0)$ with reward $b = 7$. The empirical results are illustrated in Figure 1. We can find that the empirical results agree with our theoretical derivation for all four iterations over the four sets of initialization. The learning curves of the policy $p$ and $q$ are included in Figure 5 in Appendix D. These empirical results corroborate our discussion about the limitation of $f$-divergence formulation.

## 5.2 Evaluation of TVPO

We compare TVPO with four baselines: IQL (Tan, 1993), IPPO (de Witt et al., 2020), I2Q (Jiang & Lu, 2022), and DPO (Su & Lu, 2022b). IQL is a basic value-based algorithm for decentralized learning. IPPO is a basic policy-based algorithm for decentralized learning. Both IQL and IPPO do not have convergence guarantee, to the best of our knowledge. DPO and I2Q are the recent policy-based algorithm and value-based algorithm respectively, and both of them have been proved to have convergence guarantee. In our experiments, all the algorithms use the independent parameter to agree with the fully decentralized setting, and parameter sharing is banned. More details about the experiment settings and hyperparameters are available in Appendix B and C.

**MPE** is a popular environment in cooperative MARL. MPE is a 2D environment and the objects are either agents or landmarks. Landmark is a part of the environment, while agents can move in any direction. With the relation between agents and landmarks, we can design different tasks. We use the discrete action space version of MPE and the agents can accelerate or decelerate in the direction of the x-axis or y-axis. We choose MPE for its partial observability.

The empirical results in MPE are illustrated in Figure 2. We find that TVPO obtains the best performance in all three tasks. In this environment, the policy-based algorithms, TVPO, DPO, and IPPO, outperform the value-based algorithms, IQL and I2Q. I2Q has a better performance than IQL in all three tasks.

**SMAC** is a partially observable and high-dimensional environment that has been used in many cooperative MARL studies. We select five maps in SMAC, 2s3z, 8m, 3s5z, MMM2 and 27m_vs_30m for our experiments. These maps cover all three difficulty levels in SMAC: 2s3z and 8m are easy maps; 3s5z is a hard map; MMM2 and 27m_vs_30m are super-hard maps.

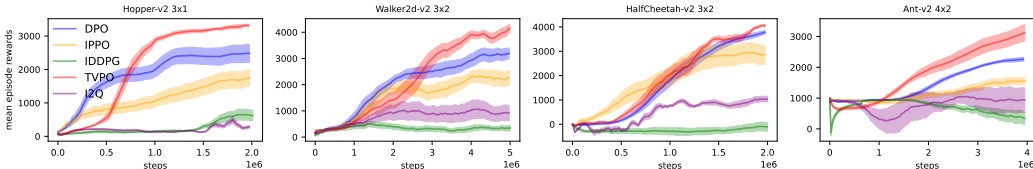

Figure 3: Learning curves of TVPO compared with IQL, IPPO, I2Q, and DPO on the maps 2s3z, 3s5z, 8m, MMM2 and 27m_vs_30m in SMAC.

Figure 4: Learning curves of TVPO compared with IDDPG, IPPO, I2Q, and DPO in 3-agent Hopper, 3-agent Walker2d, 3-agent HalfCheetah and 4-agent Ant in multi-agent MuJoCo.

We show the empirical results of these algorithms in Figure 3. In the super-hard maps MMM2 and 27m_vs_30m, all the algorithms can hardly win, so we use episode rewards as the evaluation metric to show the difference more clearly. As illustrated in Figure 3, TVPO has the best performance in all four maps. The performance of DPO and TVPO is similar in the map 8m, and the reason may be that 8m is very easy and both of them can obtain nearly 100% win rates within one million steps. In the other four maps, the differences between TVPO and DPO are more clear.

**Multi-Agent MuJoCo** is a robotic locomotion control environment for multi-agent settings, which is built upon single-agent MuJoCo (Todorov et al., 2012). In multi-agent MuJoCo, each agent controls one part of a robot to carry out different tasks. We choose this environment for the reason of continuous state and action spaces. We use independent DDPG (Lillicrap et al., 2016) (IDDPG) to replace IQL for continuous action spaces. As discussed in Section 4.3, we use Hellinger distance to replace total variation distance for continuous action space in TVPO. We select 4 tasks for our experiments: 3-agent Hopper, 3-agent HalfCheetah, 3-agent Walker2d, and 4-agent Ant. In all these tasks, we set agent_obsk=2.

The learning curves of the multi-agent MuJoCo tasks are illustrated in Figure 4. We can find that TVPO substantially outperforms the baselines except in 3-agent HalfCheetah, where DPO obtains similar performance to TVPO. The difference between the performance of the value-based algorithms and the policy-based algorithms is larger in multi-agent MuJoCo compared with MPE and SMAC. The reason may be that the continuous action space in fully decentralized learning brings more difficulty in training for the value-based algorithms.

In all three environments, TVPO obtains the best performance in all the evaluated tasks compared with the four baselines, and the differences between TVPO and the other baselines are obvious in most tasks. The performance of TVPO empirically verifies our discussion about the convergence guarantee of TVPO and the effectiveness of TVPO. Among the baselines, DPO has the closest performance to TVPO and their difference in performance empirically verifies our discussion about the advantage of TVPO over DPO.

## 6 CONCLUSION

In this paper, we propose $f$-divergence policy optimization, a general formulation of independent policy optimization in cooperative multi-agent reinforcement learning, and analyze the policy iteration of such a formulation. We discuss the limitation of this formulation, i.e., convergence to only sub-optimal policy, and verify it by the empirical results in a two-player matrix game. Based on $f$-divergence policy optimization, we propose a novel independent learning algorithm, TVPO, and prove its convergence in fully decentralized learning. Empirically, we evaluate TVPO against four baselines: IQL, IPPO, I2Q, and DPO in three environments including MPE, SMAC, and multi-agent MuJoCo. The empirical results show that TVPO outperforms all the baselines, which verifies the effectiveness of TVPO.

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

# Appendices

## A  PROOFS

### A.1  PROOF OF LEMMA 1

*Proof.* The Lagrangian function of (4) is as follows:

$$
L = \sum_{a_i} \pi^i(a_i|s) Q_i^{\boldsymbol{\pi}_{\text{old}}}(s, a_i) - \omega \sum_{a_i} \pi_{\text{old}}^i(a_i|s) f\left(\frac{\pi^i(a_i|s)}{\pi_{\text{old}}^i(a_i|s)}\right)
$$
$$
+ \lambda_s \left(\sum_{a_i} \pi^i(a_i|s) - 1\right) + \sum_{a_i} \beta(a_i|s)\pi^i(a_i|s),
$$

where $\lambda_s$ and $\beta(a_i|s)$ are the Lagrangian multiplier.

Then by the KKT condition we have

$$
\frac{\partial L}{\partial \pi^i(a_i|s)} = Q_i^{\boldsymbol{\pi}_{\text{old}}}(s, a_i) - \omega f'\left(\frac{\pi^i(a_i|s)}{\pi_{\text{old}}^i(a_i|s)}\right) + \lambda_s + \beta(a_i|s) = 0,
$$

so we can resolve $\pi^i(a_i|s)$ as

$$
\frac{\pi^i(a_i|s)}{\pi_{\text{old}}^i(a_i|s)} = g\left(\frac{Q_i^{\boldsymbol{\pi}_{\text{old}}}(s, a_i) + \lambda_s + \beta(a_i|s)}{\omega}\right) \tag{18}
$$

From the complementary slackness we know that $\beta(a_i|s)\pi^i(a_i|s) = 0$, so we can rewrite (18) as

$$
\frac{\pi^i(a_i|s)}{\pi_{\text{old}}^i(a_i|s)} = \max\left\{g\left(\frac{Q_i^{\boldsymbol{\pi}_{\text{old}}}(s, a_i) + \lambda_s}{\omega}\right), 0\right\}, \tag{19}
$$

$$
\pi^i(a_i|s) = \max\left\{\pi_{\text{old}}^i(a_i|s) g\left(\frac{Q_i^{\boldsymbol{\pi}_{\text{old}}}(s, a_i) + \lambda_s}{\omega}\right), 0\right\}. \tag{20}
$$

$\square$

### A.2  PROOF OF PROPOSITION 1

*Proof.* To discuss the monotonicity of the policies $p_t$ and $q_t$, let $Q_t^A(0)$ and $Q_t^A(1)$ represent the expected reward Alice will obtain by taking action $u_A^0$ and $u_A^1$ respectively. Simlilarly, we can also define $Q_t^B(0)$ and $Q_t^B(1)$ for Bob.

From the definition, we have $Q_t^A(0) = q_t \cdot a + (1 - q_t) \cdot b = b + (a - b)q_t$. Similarly we can obtain that $Q_t^A(1) = d + (c - d)q_t$, $Q_t^B(0) = c + (a - c)p_t$ and $Q_t^B(1) = d + (b - d)p_t$.

Combining (20) with the condition $g(x) \geq 0$, then we have

$$
p_{t+1} = p_t g\left(\frac{(a - b)q_t + b + \lambda_t^A}{\omega}\right), \ 1 - p_{t+1} = (1 - p_t) g\left(\frac{(c - d)q_t + d + \lambda_t^A}{\omega}\right)
$$

$$
\Rightarrow \quad \frac{1}{p_{t+1}} - 1 = \left(\frac{1}{p_t} - 1\right) \frac{g\left(\frac{(c-d)q_t + d + \lambda_t^A}{\omega}\right)}{g\left(\frac{(a-b)q_t + b + \lambda_t^A}{\omega}\right)}. \tag{21}
$$

From (21) we can find that

$$
p_{t+1} < p_t \quad \Leftrightarrow \quad \frac{g\left(\frac{(c-d)q_t + d + \lambda_t^A}{\omega}\right)}{g\left(\frac{(a-b)q_t + b + \lambda_t^A}{\omega}\right)} > 1
$$

$$
\Leftrightarrow \quad (c - d)q_t + d > (a - b)q_t + b \tag{22}
$$
$$
\Leftrightarrow \quad (b + c - a - d)q_t > b - d
$$
$$
\Leftrightarrow \quad q_t > \hat{q}.
$$

The critical step (22) is from the combination of the condition $g(x) \geq 0$ and the property $g(x)$ is non-decreasing.

Similarly we can obtain that $p_t > \hat{p} \Rightarrow q_{t+1} < q_t$; $p_t < \hat{p} \Rightarrow q_{t+1} > q_t$; $q_t > \hat{q} \Rightarrow p_{t+1} < p_t$; and $q_t < \hat{q} \Rightarrow p_{t+1} > p_t$. $\qquad\square$

### A.3 PROOF OF COROLLARY 1

*Proof.* From the iteration of $\{p_t\}$ we have

$$\frac{p_{t+1}}{1 - p_{t+1}} = \frac{p_t}{1 - p_t} \frac{g\left(\frac{(a-b)q_t + b + \lambda_t^A}{\omega}\right)}{g\left(\frac{(c-d)q_t + d + \lambda_t^A}{\omega}\right)}. \tag{23}$$

Let $t \to \infty$ in both side of (23), we know that

$$\frac{p^*}{1 - p^*}\left(\frac{g\left(\frac{(a-b)q^* + b + \lambda_*^A}{\omega}\right)}{g\left(\frac{(c-d)q^* + d + \lambda_*^A}{\omega}\right)} - 1\right) = 0. \tag{24}$$

As $q^* > \hat{q}$, we know that $\dfrac{g\left(\frac{(a-b)q^* + b + \lambda_*^A}{\omega}\right)}{g\left(\frac{(c-d)q^* + d + \lambda_*^A}{\omega}\right)} < 1$. So we can rewrite (24) as $\frac{p^*}{1-p^*} = 0$ and resolve $p^* = 0$.

As for $q^*$, we can follow a similar idea. From the iteration of $\{q_t\}$ we have

$$\frac{1}{q_{t+1}} - 1 = (\frac{1}{q_t} - 1)\frac{g\left(\frac{(b-d)p_t + d + \lambda_t^B}{\omega}\right)}{g\left(\frac{(a-c)p_t + c + \lambda_t^B}{\omega}\right)}. \tag{25}$$

Let $t \to \infty$ in both side of (25) , we know that

$$\frac{1 - q^*}{q^*}\left(\frac{g\left(\frac{(b-d)p^* + d + \lambda_*^B}{\omega}\right)}{g\left(\frac{(a-c)p^* + c + \lambda_*^B}{\omega}\right)} - 1\right) = 0. \tag{26}$$

As $p^* < \hat{p}$, we know that $\dfrac{g\left(\frac{(b-d)p^* + d + \lambda_*^B}{\omega}\right)}{g\left(\frac{(a-c)p^* + c + \lambda_*^B}{\omega}\right)} < 1$. Then we can rewrite (26) as $\frac{1-q^*}{q^*} = 0$ and obtain $q^* = 1$. $\qquad\square$

### A.4 PROOF OF LEMMA 2

*Proof.* For any fixed $i$, consider the following difference

$$\left| \sum_{\boldsymbol{a}} \boldsymbol{\pi}_{\text{new}}(\boldsymbol{a}|s) Q^{\boldsymbol{\pi}}(s, \boldsymbol{a}) - \sum_{a_i} \pi_{\text{new}}^i(a_i|s) \sum_{a_{-i}} \pi_{\text{old}}^{-i}(a_{-i}|s) Q^{\boldsymbol{\pi}}(s, a_i, a_{-i}) \right|$$

$$= \left| \sum_{a_i} \pi_{\text{new}}^i(a_i|s) \sum_{a_{-i}} \left( \pi_{\text{new}}^{-i}(a_{-i}|s) - \pi_{\text{old}}^{-i}(a_{-i}|s) \right) Q^{\boldsymbol{\pi}}(s, a_i, a_{-i}) \right| \tag{27}$$

$$\leq \sum_{a_i} \pi_{\text{new}}^i(a_i|s) \sum_{a_{-i}} \left| \pi_{\text{new}}^{-i}(a_{-i}|s) - \pi_{\text{old}}^{-i}(a_{-i}|s) \right| \left| Q^{\boldsymbol{\pi}}(s, a_i, a_{-i}) \right| \tag{28}$$

$$\leq \frac{M}{2} \sum_{a_i} \pi_{\text{new}}^i(a_i|s) \sum_{a_{-i}} \left| \pi_{\text{new}}^{-i}(a_{-i}|s) - \pi_{\text{old}}^{-i}(a_{-i}|s) \right| \tag{29}$$

$$= \frac{M}{2} \sum_{a_{-i}} \left| \pi_{\text{new}}^{-i}(a_{-i}|s) - \pi_{\text{old}}^{-i}(a_{-i}|s) \right| \tag{30}$$

$$= \frac{M}{2} \sum_{a_{-i}} \left| \sum_{k=1, k \neq i}^{N} \pi_{\text{new}}^{1:k-1}(a_{1:k-1}|s) \pi_{\text{old}}^{k:N}(a_{k:N}|s) - \pi_{\text{new}}^{1:k}(a_{1:k}|s) \pi_{\text{old}}^{k+1 \sim N}(a_{k+1:N}|s) \right| \tag{31}$$

$$\leq \frac{M}{2} \sum_{a_{-i}} \sum_{k=1, k \neq i}^{N} \left| \pi_{\text{new}}^{1:k-1}(a_{1:k-1}|s) \pi_{\text{old}}^{k:N}(a_{k:N}|s) - \pi_{\text{new}}^{1:k}(a_{1:k}|s) \pi_{\text{old}}^{k+1 \sim N}(a_{k+1:N}|s) \right| \tag{32}$$

$$= \frac{M}{2} \sum_{k=1, k \neq i}^{N} \sum_{a_k} \left| \pi_{\text{new}}^k(a_k|s) - \pi_{\text{old}}^k(a_k|s) \right| \tag{33}$$

$$= M \sum_{k=1, k \neq i}^{N} D_{\text{TV}} \left( \pi_{\text{new}}^k(\cdot|s) \| \pi_{\text{old}}^k(\cdot|s) \right) \tag{34}$$

where $\pi_{\text{new}}^{1:k-1}$ denotes $\pi_{\text{new}}^1 \times \pi_{\text{new}}^2 \times \cdots \pi_{\text{new}}^{k-1}$ and $\pi_{\text{new}}^i$ will be skipped if involved, and $a_{1:k-1}$ has similar meanings as $a_{1:k-1} = a_1 \times a_2 \times \cdots a_{k-1}$. In (28) and (32), we use the triangle inequality of the absolute value. In (29), we use the property $Q^{\boldsymbol{\pi}}(s, \boldsymbol{a}) \leq \frac{r_{\max}}{1-\gamma} = \frac{M}{2}$ from the definition of Q-function. In (31), we insert $N - 1$ terms between $\pi_{\text{new}}^{-i}(a_{-i}|s)$ and $\pi_{\text{old}}^{-i}(a_{-i}|s)$ to make sure the adjacent two terms are only different in one individual policy.

By rewriting the conclusion above, for any agent $i$, we have

$$\sum_{\boldsymbol{a}} \boldsymbol{\pi}_{\text{new}}(\boldsymbol{a}|s) Q^{\boldsymbol{\pi}}(s, \boldsymbol{a}) \geq \sum_{a_i} \pi_{\text{new}}^i(a_i|s) \sum_{a_{-i}} \pi_{\text{old}}^{-i}(a_{-i}|s) Q^{\boldsymbol{\pi}}(s, a_i, a_{-i})$$

$$- M \sum_{k=1, k \neq i}^{N} D_{\text{TV}} \left( \pi_{\text{new}}^k(\cdot|s) \| \pi_{\text{old}}^k(\cdot|s) \right). \tag{35}$$

Then, by applying (35) to $i = 1, 2, \cdots, N$ and add all these $N$ inequalities together, we have

$$\sum_{\boldsymbol{a}} \boldsymbol{\pi}_{\text{new}}(\boldsymbol{a}|s) Q^{\boldsymbol{\pi}}(s, \boldsymbol{a}) \geq \frac{1}{N} \sum_{i=1}^{N} \sum_{a_i} \pi_{\text{new}}^i(a_i|s) \sum_{a_{-i}} \pi_{\text{old}}^{-i}(a_{-i}|s) Q^{\boldsymbol{\pi}}(s, a_i, a_{-i})$$

$$- \frac{(N-1)M}{N} \sum_{i=1}^{N} D_{\text{TV}} \left( \pi_{\text{new}}^i(\cdot|s) \| \pi_{\text{old}}^i(\cdot|s) \right).$$

$$\square$$

### A.5 PROOF OF PROPOSITION 2

*Proof.* By the definition of $V_{\boldsymbol{\rho}}^{\boldsymbol{\pi}_{\text{old}}}$ we have

$$V_{\boldsymbol{\rho}}^{\boldsymbol{\pi}_{\text{old}}}(s) = \frac{1}{N}\sum_i \sum_{a_i} \pi_{\text{old}}^i(a_i|s) \sum_{a_{-i}} \rho^{-i}(a_{-i}|s) Q_{\boldsymbol{\rho}}^{\boldsymbol{\pi}_{\text{old}}}(s, a_i, a_{-i}) - \omega \sum_i D_f\left(\pi_{\text{old}}^i(\cdot|s)\|\rho^i(\cdot|s)\right)$$

$$\leq \frac{1}{N}\sum_i \sum_{a_i} \pi_{\text{new}}^i(a_i|s) \sum_{a_{-i}} \rho^{-i}(a_{-i}|s) Q_{\boldsymbol{\rho}}^{\boldsymbol{\pi}_{\text{old}}}(s, a_i, a_{-i}) - \omega \sum_i D_f\left(\pi_{\text{new}}^i(\cdot|s)\|\rho^i(\cdot|s)\right) \quad (36)$$

$$= \frac{1}{N}\sum_i \sum_{a_i} \pi_{\text{new}}^i(a_i|s) \sum_{a_{-i}} \rho^{-i}(a_{-i}|s) \left(r(s, a_i, a_{-i}) + \gamma\mathbb{E}\left[V_{\boldsymbol{\rho}}^{\boldsymbol{\pi}_{\text{old}}}(s')\right]\right)$$

$$- \omega \sum_i D_f\left(\pi_{\text{new}}^i(\cdot|s)\|\rho^i(\cdot|s)\right) \quad (37)$$

$$\leq \cdots \quad (\text{expand } V_{\boldsymbol{\rho}}^{\boldsymbol{\pi}_{\text{old}}}(s') \text{ and repeat replacing } \pi_{\text{old}}^i \text{ with } \pi_{\text{new}}^i) \quad (38)$$

$$\leq V_{\boldsymbol{\rho}}^{\boldsymbol{\pi}_{\text{new}}}(s). \quad (39)$$

In (36), we use the definition of $\pi_{\text{new}}^i$ in (11). (37) is from the definition of $Q_{\boldsymbol{\rho}}^{\boldsymbol{\pi}_{\text{old}}}(s, a_i, a_{-i})$. In (38), we repeatedly expand $V_{\boldsymbol{\rho}}^{\boldsymbol{\pi}_{\text{old}}}$ according to its definition and replace $\pi_{\text{old}}^i$ with $\pi_{\text{new}}^i$ by the optimality of $\pi_{\text{new}}^i$ like what we have done in (36). After we replace all $\pi_{\text{old}}^i$ with $\pi_{\text{new}}^i$, then we obtain $V_{\boldsymbol{\rho}}^{\boldsymbol{\pi}_{\text{new}}}(s)$ according to the definition of $V_{\boldsymbol{\rho}}^{\boldsymbol{\pi}_{\text{new}}}(s)$ in (39).

With the result $V_{\boldsymbol{\rho}}^{\boldsymbol{\pi}_{\text{old}}}(s) \leq V_{\boldsymbol{\rho}}^{\boldsymbol{\pi}_{\text{new}}}(s)$, we know $Q_{\boldsymbol{\rho}}^{\boldsymbol{\pi}_{\text{old}}}(s, \boldsymbol{a}) = r(s, \boldsymbol{a}) + \gamma\mathbb{E}[V_{\boldsymbol{\rho}}^{\boldsymbol{\pi}_{\text{old}}}(s')] \leq r(s, \boldsymbol{a}) + \gamma\mathbb{E}[V_{\boldsymbol{\rho}}^{\boldsymbol{\pi}_{\text{new}}}(s')] = Q_{\boldsymbol{\rho}}^{\boldsymbol{\pi}_{\text{new}}}(s, \boldsymbol{a})$. □

### A.6 PROOF OF THEOREM 1

*Proof.* From the Proposition 2, we know $V_{\boldsymbol{\pi}_t}^{\boldsymbol{\pi}_{t+1}}(s) \geq V^{\boldsymbol{\pi}_t}(s)$. Thus, we just need to prove $V^{\boldsymbol{\pi}_t}(s) \geq V_{\boldsymbol{\pi}_{t-1}}^{\boldsymbol{\pi}_t}(s)$.

From the definition of $V^{\boldsymbol{\pi}_t}(s)$ we have

$$V^{\boldsymbol{\pi}_t}(s) = \sum_{\boldsymbol{a}} \boldsymbol{\pi}_t(\boldsymbol{a}|s) Q^{\boldsymbol{\pi}_t}(s, \boldsymbol{a})$$

$$\geq \frac{1}{N}\sum_{i=1}^N \sum_{a_i} \pi_t^i(a_i|s) \sum_{a_{-i}} \pi_{t-1}^{-i}(a_{-i}|s) Q^{\boldsymbol{\pi}_t}(s, a_i, a_{-i})$$

$$- \omega \sum_{i=1}^N D_{\text{TV}}\left(\pi_t^i(\cdot|s)\|\pi_{t-1}^i(\cdot|s)\right) \quad (40)$$

$$= \frac{1}{N}\sum_{i=1}^N \sum_{a_i} \pi_t^i(a_i|s) \sum_{a_{-i}} \pi_{t-1}^{-i}(a_{-i}|s) \left(r(s, a_i, a_{-i}) + \gamma\mathbb{E}[V^{\boldsymbol{\pi}_t}(s')]\right)$$

$$- \omega \sum_{i=1}^N D_{\text{TV}}\left(\pi_t^i(\cdot|s)\|\pi_{t-1}^i(\cdot|s)\right) \quad (41)$$

$$\geq \cdots \quad (\text{expand } V^{\boldsymbol{\pi}_t}(s') \text{ and repeat replacing } \pi_t^{-i} \text{ with } \pi_{t-1}^{-i}) \quad (42)$$

$$\geq V_{\boldsymbol{\pi}_{t-1}}^{\boldsymbol{\pi}_t}(s). \quad (43)$$

(40) is from Lemma 2, and (41) is from the definition of $Q^{\boldsymbol{\pi}_t}(s, a_i, a_{-i})$. In (42), we repeatedly expand $V^{\boldsymbol{\pi}_t}$ and replace the $\pi_t^{-i}$ with $\pi_{t-1}^{-i}$ by Lemma 2 like what we have done in (40). After we replace all $\pi_t^{-i}$ with $\pi_{t-1}^{-i}$, then we obtain $V_{\boldsymbol{\pi}_{t-1}}^{\boldsymbol{\pi}_t}(s)$ in (43) according to the definition of $V_{\boldsymbol{\pi}_{t-1}}^{\boldsymbol{\pi}_t}(s)$.

From the inequalities $V_{\boldsymbol{\pi}_t}^{\boldsymbol{\pi}_{t+1}}(s) \geq V^{\boldsymbol{\pi}_t}(s) \geq V_{\boldsymbol{\pi}_{t-1}}^{\boldsymbol{\pi}_t}(s) \geq V^{\boldsymbol{\pi}_{t-1}}(s)$, we know that the sequence $\{V^{\boldsymbol{\pi}_t}\}$ improves monotonically. Combining with the condition that the sequence $\{V^{\boldsymbol{\pi}_t}\}$ is bounded, we know that $\{V^{\boldsymbol{\pi}_t}\}$ will converge to $V^*$. According to the definition, the sequence $\{Q^{\boldsymbol{\pi}_t}\}$ and $\{\boldsymbol{\pi}_t\}$

will also converge to $Q^*$ and $\boldsymbol{\pi}_*$ respectively, where $\boldsymbol{\pi}_*$ satisfies the following fixed-point equation:

$$\pi_*^i = \arg\max_{\pi^i} \sum_{a_i} \pi^i(a_i|s) \sum_{a_{-i}} \pi_*^{-i}(a_{-i}|s) Q^*(s, a_i, a_{-i}) - \omega D_{\text{TV}}\left(\pi^i(\cdot|s) \| \pi_*^i(\cdot|s)\right).$$

□

## A.7 Proof of $D_{\text{TV}}(p\|q) \leq D_{\text{H}}(p\|q)$

*Proof.*

$$
\begin{aligned}
D_{\text{TV}}^2(p\|q) &= \frac{1}{4}\left(\sum_i |p_i - q_i|\right)^2 = \frac{1}{4}\left(\sum_i |\sqrt{p_i} - \sqrt{q_i}|\,|\sqrt{p_i} + \sqrt{q_i}|\right)^2 \\
&\leq \frac{1}{4}\left(\sum_i |\sqrt{p_i} - \sqrt{q_i}|^2\right)\left(\sum_i |\sqrt{p_i} + \sqrt{q_i}|^2\right) \quad \text{(Cauchy–Schwarz inequality)} \\
&= \frac{1}{4}D_{\text{H}}^2(p\|q)\left(2 + 2\sum_i \sqrt{p_i q_i}\right) \\
&\leq D_{\text{H}}^2(p\|q).
\end{aligned}
$$

□

# B EXPERIMENTAL SETTINGS

## B.1 MPE

The three tasks are built on the origin MPE (Lowe et al., 2017) (MIT license) and are originally used in Agarwal et al. (2020) (MIT license). The objective in these three tasks are listed as follows:

- **Simple Spread:** There are $N$ agents who need to occupy the locations of $N$ landmarks.
- **Line Control:** There are $N$ agents who need to line up between 2 landmarks.
- **Circle Control:** There are $N$ agents who need to form a circle around a landmark.

The reward in these tasks is the distance between all the agents and their target locations. We set the number of agents $N = 10$ for these three tasks in our experiment.

## B.2 MULTI-AGENT MUJOCO

Multi-agent MuJoCo (Peng et al., 2021) (Apache-2.0 license) is a robotic locomotion task with continuous action space for multi-agent settings. The robot could be divided into several parts and each part contains several joints. Agents in this environment control a part of the robot which could be different varieties. So the type of the robot and the assignment of the joints decide a task. For example, the task 'HalfCheetah-3×2' means dividing the robot 'HalfCheetah' into three parts for three agents and each part contains 2 joints.

The details about our experiment settings in multi-agent Mujoco are listed in Table 2. The configuration defines the number of agents and the joints of each agent. The 'agent obsk' defines the number of nearest agents an agent can observe.

Table 2: The task settings of multi-agent MuJoCo

| task | configuration | agent obsk |
|---|---|---|
| HalfCheetah | 3×2 | 2 |
| Hopper | 3×1 | 2 |
| Walker2d | 3×2 | 2 |
| Ant | 4×2 | 2 |

## C   TRAINING DETAILS

Our code of IPPO is based on the open-source code[1] of MAPPO (Yu et al., 2021) (MIT license). We modify the code for individual parameters and ban the tricks used by MAPPO for SMAC. The network architectures and base hyperparameters of TVPO, DPO and IPPO are the same for all the tasks in all the environments. We use 3-layer MLPs for the actor and the critic and use ReLU as non-linearities. The number of the hidden units of the MLP is 128. We train all the networks with an Adam optimizer. The learning rates of the actor and critic are both 5e-4. The number of epochs for every batch of samples is 15 which is the recommended value in Yu et al. (2021). For IPPO, the clip parameter is 0.2 which is the same as Schulman et al. (2017). For DPO, the hyperparameter is set as the original paper (Su & Lu, 2022b) recommends. Our code of IQL is based on the open-source code[2] PyMARL (Apache-2.0 license) and we modify the code for individual parameters. The default architecture in PyMARL is RNN so we just follow it and the number of the hidden units is 128. The learning rate of IQL is also 5e-4. The architectures of the actor and critic of IDDPG are 3-layer MLPs. The learning rates of the actor and critic are both 5e-4. Our code of I2Q is from the open source code[3] of the original paper (Jiang & Lu, 2022). We keep the hyperparameter of I2Q the same as the default value of the open-source code in our experiments.

Table 3: Hyperparameters for all the experiments

| hyperparameter | value |
|---|---|
| MLP layers | 3 |
| hidden size | 128 |
| non-linear | ReLU |
| optimizer | Adam |
| actor_lr | 5e-4 |
| critic_lr | 5e-4 |
| numbers of epochs | 15 |
| initial $\beta^i$ | 0.01 |
| $\delta$ | 1.5 |
| $\omega$ | 2 |
| $d_{\text{target}}$ | 0.001 |
| clip parameter for IPPO | 0.2 |

The version of the game StarCraft2 in SMAC is 4.10 for our experiments in all the SMAC tasks. We set the episode length of all the multi-agent MuJoCo tasks as 1000 in all of our multi-agent MuJoCo experiments. We perform the whole experiment with a total of four NVIDIA A100 GPUs. We have summarized the hyperparameters in Table 3.

## D   ADDITIONAL EMPIRICAL RESULTS

Figure 5 illustrates the learning curve of the policy $p$ and $q$ in the matrix game of KL-iteration, TV-iteration, $\chi^2$-iteration, and H-iteration over four different sets of initialization. We can the policies of all four kinds of iterations converge.

## E   DISCUSSION

Before proposing the $f$-divergence formulation, we have studied another formulation. This formulation follow the idea of the entropy regularization and the extra term is only related to the policy $pi^i$ instead of the divergence between $pi^i$ and $pi^i_{\text{old}}$. We refer to this approach as the unary formulation. Though we discovered that the unary formulation has more significant drawbacks, the properties of the unary formulation inspire us in the proof of TVPO. So we would like to provide the properties and some empirical results of the unary formulation here for discussion.

---

[1] https://github.com/marlbenchmark/on-policy
[2] https://github.com/oxwhirl/pymarl
[3] https://github.com/jiechuanjiang/I2Q

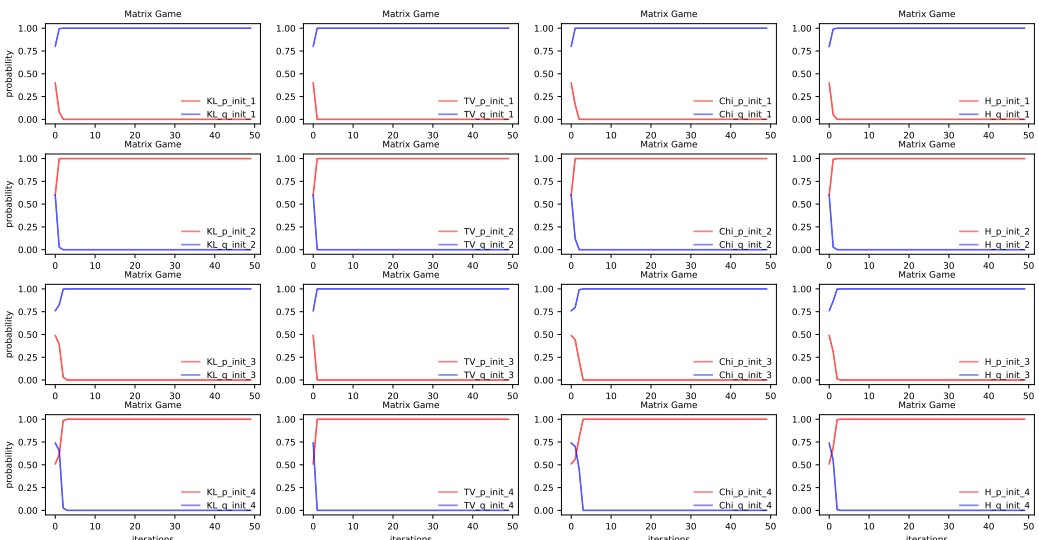

Figure 5: Learning curves of the policy $p$ and $q$ in the matrix game of KL-iteration, TV-iteration, $\chi^2$-iteration, and H-iteration over four different sets of initialization. Each row corresponds to one set of initialization and each column corresponds to one type of iteration.

### E.1 UNARY FORMULATION

The unary formulation is

$$\pi_{\text{new}}^i = \arg\max_{\pi^i} \sum_{a_i} \pi^i(a_i|s) Q_i^{\boldsymbol{\pi}_{\text{old}}}(s, a_i) + \omega \sum_{a_i} \pi^i(a_i|s) \phi\left(\pi^i(a_i|s)\right). \tag{44}$$

This formulation (44) follows the idea of Yang et al. (2019) which discusses the regularization algorithm in single-agent RL. From the perspective of regularization, the update rule (44) can be seen as optimizing the regularized objective $J_\phi^i(\boldsymbol{\pi}) = \mathbb{E}\left[\sum_t \gamma^t \left(r_i(s, a_i) + \omega\phi\left(\pi^i(a_i|s)\right)\right)\right]$, where $r_i(s, a_i) = \mathbb{E}_{\pi^{-i}}[r(s, a_i, a_{-i})]$. The choice of $\phi$ is flexible, *e.g.,* $\phi(x) = -\log x$ corresponds to entropy regularization and independent SAC (Haarnoja et al., 2018); $\phi(x) = 0$ means (44) degenerates to independent Q-learning (Tan, 1993); Moreover, there are many other options for $\phi$ corresponding to different regularization (Yang et al., 2019). So we take (44) as the general unary formulation of independent learning, where the 'unary' means the additional terms $\sum_{a_i} \pi^i(a_i|s)\phi\left(\pi^i(a_i|s)\right)$ is only about one policy $\pi^i$.

For further discussion of (44), we can utilize the conclusion in Yang et al. (2019) as the following lemma.

**Lemma 3.** *If $\phi(x)$ in $(0, 1]$ and satisfies the following conditions: (1) $\phi(x)$ is non-increasing; (2) $\phi(1) = 0$; (3) $\phi(x)$ is differentiable; (4) $f_\phi(x) = x\phi(x)$ is strictly concave, then we have that $g_\phi(x) = (f_\phi')^{-1}(x)$ exists and $g_\phi(x)$ is decreasing. Moreover, the solution to the optimization objective (44) can be described with $g_\phi(x)$ as follows:*

$$\pi_{\text{new}}^i(a_i|s) = \max\{g_\phi\left(\frac{\lambda_s - Q_i^{\boldsymbol{\pi}_{\text{old}}}(s, a_i)}{\omega}\right), 0\}, \tag{45}$$

*where $\lambda_s$ satisfies $\sum_{a_i} \max\{g_\phi\left(\frac{\lambda_s - Q_i^{\boldsymbol{\pi}_{\text{old}}}(s, a_i)}{\omega}\right), 0\} = 0$.*

Though it seems that $\phi(x)$ needs to satisfy four conditions, actually $\phi(x) = -\log x$ for Shannon entropy and $\phi(x) = \frac{k}{q-1}(1 - x^{q-1})$ for Tsallis entropy are still qualified.

However, unlike the single-agent setting, the update rule in Lemma 3 may result in the convergence to sub-optimal policy or even oscillations in policy in fully decentralized MARL.

We further discuss (44) in the two-player matrix game and have the following proposition.

**Proposition 3.** *Suppose that $g_\phi(x) \geq 0$ and $g_\phi(x)$ is continuously differentiable. If the payoff matrix of the two-player matrix game satisfies $b + c < a + d$, and two agents Alice and Bob update their policies with policy iteration as*

$$\pi_{t+1}^i = \arg\max_{\pi^i} \sum_{a_i} \pi^i(a_i|s)Q_i^{\pi_t}(s, a_i) + \omega \sum_{a_i} \pi^i(a_i|s)\phi\left(\pi^i(a_i|s)\right), \tag{46}$$

*then we have (1) $p_t > p_{t-1} \Rightarrow q_{t+1} > q_t$; (2) $p_t < p_{t-1} \Rightarrow q_{t+1} < q_t$; (3) $q_t > q_{t-1} \Rightarrow p_{t+1} > p_t$; (4) $q_t < q_{t-1} \Rightarrow p_{t+1} < p_t$.*

*Proof.* To discuss the monotonicity of the policies $p_t$ and $q_t$, we need the solution in Lemma 3. Before applying the update rule (45), we need to calculate the decentralized critic given $p_t$ and $q_t$. Let $Q_t^A(0)$ and $Q_t^A(1)$ represent the expected reward Alice will obtain by taking action $u_A^0$ and $u_A^1$ respectively. We can also define $Q_t^B(0)$ and $Q_t^B(1)$ for Bob.

From the definition, we have $Q_t^A(0) = q_t \cdot a + (1 - q_t) \cdot b = b + (a - b)q_t$. Similarly we could obtain that $Q_t^A(1) = d + (c - d)q_t$, $Q_t^B(0) = c + (a - c)p_t$ and $Q_t^B(1) = d + (b - d)p_t$.

With (45) and the condition $g_\phi(x) \geq 0$, we have

$$p_{t+1} = g_\phi\left(\frac{\lambda_t^A - Q_t^A(0)}{\omega}\right) = g_\phi\left(\frac{(b-a)q_t + \lambda_t^A - b}{\omega}\right), \quad 1 - p_{t+1} = g_\phi\left(\frac{(d-c)q_t + \lambda_t^A - d}{\omega}\right)$$

$$g_\phi\left(\frac{(b-a)q_t + \lambda_t^A - b}{\omega}\right) + g_\phi\left(\frac{(d-c)q_t + \lambda_t^A - d}{\omega}\right) = 1$$

$$q_{t+1} = g_\phi\left(\frac{(c-a)p_t + \lambda_t^B - c}{\omega}\right), \quad 1 - q_{t+1} = g_\phi\left(\frac{(d-b)p_t + \lambda_t^B - d}{\omega}\right)$$

$$g_\phi\left(\frac{(c-a)p_t + \lambda_t^B - c}{\omega}\right) + g_\phi\left(\frac{(d-b)p_t + \lambda_t^B - d}{\omega}\right) = 1.$$

We can rewrite these equations with some simplifications as follows,

$$m_A(x) \triangleq \frac{(b-a)x + \lambda_A(x) - b}{\omega}, \quad n_A(x) \triangleq \frac{(d-c)x + \lambda_A(x) - d}{\omega}, \quad h_A(x) = g_\phi(m_A(x))$$
$$\text{where } \lambda_A(x) \text{ satisfies } g_\phi(m_A(x)) + g_\phi(n_A(x)) = 1 \tag{47}$$
$$m_B(x) \triangleq \frac{(c-a)p_t + \lambda_B(x) - c}{\omega}, \quad n_B(x) \triangleq \frac{(d-b)p_t + \lambda_B(x) - d}{\omega}, \quad h_B(x) = g_\phi(m_B(x))$$
$$\text{where } \lambda_B(x) \text{ satisfies } g_\phi(m_B(x)) + g_\phi(n_B(x)) = 1.$$

With these definitions, we know that $p_{t+1} = h_A(q_t)$, $q_{t+1} = h_B(p_t)$ and the monotonicity of $p_t$ and $q_t$ is determined by the property of function $h_A(x)$ and $h_B(x)$. By applying the chain rule to (47), we have:

$$\frac{1}{\omega}g_\phi'(m_A(x))(b - a + \lambda_A'(x)) + \frac{1}{\omega}g_\phi'(n_A(x))(d - c + \lambda_A'(x)) = 0$$
$$\Rightarrow \lambda_A'(x) = -\frac{(b-a)g_\phi'(m_A(x)) + (d-c)g_\phi'(n_A(x))}{g_\phi'(m_A(x)) + g_\phi'(n_A(x))}. \tag{48}$$

Then we have:

$$h_A'(x) = \frac{1}{\omega}g_\phi'(m_A(x))(b - a + \lambda_A'(x)) \quad \text{(Apply chain rule)} \tag{49}$$

$$= \frac{1}{\omega}(b + c - a - d)\frac{g_\phi'(n_A(x))g_\phi'(m_A(x))}{g_\phi'(m_A(x)) + g_\phi'(n_A(x))} \quad \text{(Substitute (48) for } \lambda_A'(x) \text{)}. \tag{50}$$

Let $M = b + c - a - d$ and $M' = \frac{M}{\omega}$, then $h_A'(x) = M'\frac{g_\phi'(n_A(x))g_\phi'(m_A(x))}{g_\phi'(m_A(x)) + g_\phi'(n_A(x))}$. From the condition and Lemma 3 we know that $M' < 0$ and $g_\phi(x)$ is decreasing which means $g_\phi'(x) < 0$. Combining these conditions together, we know $h_A'(x) > 0$ and $h_A(x)$ is increasing which means that $p_{t+1} = h_A(q_t)$ is increasing over $q_t$, which means that $q_t > q_{t-1} \Rightarrow p_{t+1} > p_t$ and $q_t > q_{t-1} \Rightarrow p_{t+1} > p_t$.

Similarly, we can obtain that $h'_B(x) = M' \frac{g'_\phi(n_B(x))g'_\phi(m_B(x))}{g'_\phi(m_B(x))+g'_\phi(n_B(x))} > 0$ which could lead to the result that $p_t > p_{t-1} \Rightarrow q_{t+1} > q_t$ and $p_t < p_{t-1} \Rightarrow q_{t+1} < q_t$. □

Proposition 3 actually tells us $p_{t+1} = h_A(q_t)$ is increasing over $q_t$ and $q_{t+1} = h_B(p_t)$ is increasing over $p_t$ when $M = b + c - a - d < 0$. Intuitively, we can find two typical cases for policy iterations with Proposition 3. In the first case, if in a certain iteration $t$ the conditions $p_t > p_{t-1}$ and $q_t > q_{t-1}$ are satisfied, then we know that $p_{t'+1} > p_{t'}$   $q_{t'+1} > q_{t'}$   $\forall t' \geq t$. As the sequences $\{p_t\}$ and $\{q_t\}$ are both bounded in the interval $[0, 1]$, we know that $\{p_t\}$ and $\{q_t\}$ will converge to $p^*$ and $q^*$ . The property of $p^*$ and $q^*$ is determined by $l_A(x) \triangleq h_B(h_A(x))$ and $l_B(x) \triangleq h_A(h_B(x))$ respectively as $p_{t+2} = h_B(h_A(p_t))$ and $q_{t+2} = h_A(h_B(q_t))$ and we have the following corollary.

**Corollary 2.** $|l'_A(x)| \leq M'^2 U_\phi^2$, $|l'_B(x)| \leq M'^2 U_\phi^2$, where $U_\phi$ is a constant determined by $\phi(x)$.

*Proof.* As $g'_\phi(x)$ is continuous, let $U_A^1 \triangleq \max_{x \in [0,1]} |g'_\phi(m_A(x))|$, $U_A^2 \triangleq \max_{x \in [0,1]} |g'_\phi(n_A(x))|$, $U_B^1 \triangleq \max_{x \in [0,1]} |g'_\phi(m_B(x))|$ and $U_B^2 \triangleq \max_{x \in [0,1]} |g'_\phi(n_B(x))|$. Moreover, let $U_\phi = \max\{U_A^1, U_A^2, U_B^1, U_B^2\}$, then apply the chain rule to $l'_A(x)$ and we have

$$|l'_A(x)| = |h'_B(h_A(x))h'_A(x)|$$

$$= M'^2 \frac{|g'_\phi(n_B(h_A(x)))||g'_\phi(m_B(h_A(x)))|}{|g'_\phi(m_B(h_A(x)))| + |g'_\phi(n_B(h_A(x)))|} \frac{|g'_\phi(n_A(x))||g'_\phi(m_A(x))|}{|g'_\phi(m_A(x))| + |g'_\phi(n_A(x))|} \tag{51}$$

$$= M'^2 \frac{|g'_\phi(n_B(y))||g'_\phi(m_B(y))|}{|g'_\phi(m_B(y))| + |g'_\phi(n_B(y))|} \frac{|g'_\phi(n_A(x))||g'_\phi(m_A(x))|}{|g'_\phi(m_A(x))| + |g'_\phi(n_A(x))|} \quad \text{(Let } y = h_A(x) \in [0, 1])$$

$$\leq M'^2 \frac{|g'_\phi(m_B(y))| + |g'_\phi(n_B(y))|}{2} \frac{|g'_\phi(m_A(x))| + |g'_\phi(n_A(x))|}{2} \tag{52}$$

$$\leq M'^2 U_\phi^2 \tag{53}$$

where (51) is from Proposition 3, (52) is from the AM-GM inequality $ab \leq \frac{(a+b)^2}{2}$, and (53) is from the definition of $U_\phi$. Similarly, we can obtain $|l'_B(x)| \leq M'^2 U_\phi^2$. □

Combining Corollary 2 and Banach fixed-point theorem, we can find that as $U_\phi$ is a constant, if $|M'| < \frac{1}{U_\phi}$, then we can find a constant $L$ such that $|l'_A(x)| \leq M'^2 U_\phi^2 \leq L < 1$, which means that the iteration $p_{t+1} = l_A(p_t)$ is a contraction and $p^*$ is the unique fixed-point of $l_A$. This conclusion can be seen as that a smaller $|M'|$ corresponds to a larger probability of convergence. In this convergence case, the converged policies $p^*$ and $q^*$ are usually not the optimal policy as the optimal policy is deterministic, which can be seen in our empirical results.

In the second case, which may be more general, in iteration $t$, $(p_t - p_{t-1})(q_t - q_{t-1}) < 0$, which means $p_t > p_{t-1}$ and $q_t < q_{t-1}$ or $p_t < p_{t-1}$ and $q_t > q_{t-1}$. Without loss of generality, we assume $p_t > p_{t-1}$ and $q_t < q_{t-1}$, then we know $p_{t+1} < p_t$ and $q_{t+1} < q_t$ from Proposition 3. By induction we can find that for any $t' \geq t$, the sequence $\{p_{t'}\}$ and $\{q_{t'}\}$ will increase and decrease alternatively, which means that the policies may not converge but oscillate. We will show this in our experiments. As the unary formulation may result in policy oscillation, we would like to find other formulations for fully decentralized MARL.

### E.2 VERIFICATION FOR UNARY FORMULATION

In this section, we choose $\phi(x) = -\log x$ corresponding to the entropy regularization as the representation for the unary formulation. We build two cases to show the convergence to the sub-optimal policy and the policy oscillation. We choose $a = 5, b = 6, c = 3, d = 5$ as case 2 and $a = 7, b = 5, c = 4, d = 6$ as case 3. Both two cases satisfy the condition $b + c < a + d$ as discussed above. We keep $\omega = 0.1$ for all the experiments on these two matrix games. The empirical results are illustrated in Figure 6. We can find the policies $p$ and $q$ improve monotonically to the convergence $(p^*, q^*) \approx (0.773, 0.227)$ in case 2, which is a sub-optimal joint policy. However, in case 3, the policies $p$ and $q$ oscillate between 0 and 1 and do not converge. These results verify our discussion about the limitation of the unary formulation.

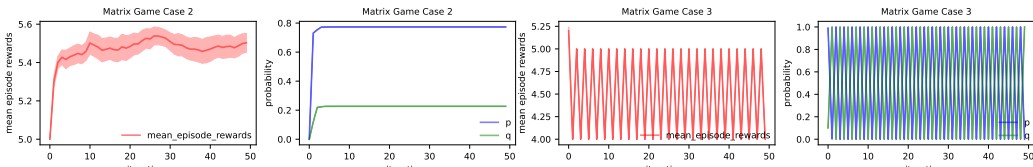

Figure 6: Learning curves of the unary formulation in two matrix game cases, where x-axis is iteration steps. The first and second figures show the performance and the policies $p$ and $q$ in the matrix game case 2 respectively. The third and fourth figures show the performance and the policies $p$ and $q$ in the matrix game case 3 respectively.

