# OpenReview forum: "A General Formulation of Independent Policy Optimization in Fully Decentralized MARL"
_ICLR.cc/2024/Conference — ICLR 2024 Conference Withdrawn Submission_

### Official Review · Reviewer_KN9g · 2023-10-30

**Soundness:** 1 poor
**Presentation:** 2 fair
**Contribution:** 1 poor
**Rating:** 1
**Confidence:** 5

**Summary:**

The paper considers the independent learning in cooperative multi-agent reinforcement learning (MARL) and proposes f-divergence policy optimization framework, which yields a practical independent learning algorithm, Total Variation Policy Optimization (TVPO). The paper shows that TVPO outperforms IQL, IPPO, I2Q and DPO.

**Strengths:**

### Idea originality & clarity
In terms of the idea, the paper is well-motivated to study the independent learning method with theoretical guarantees. The derivation presented in the paper is straightforward and and easy to follow.

**Weaknesses:**

### Technical novelty
**Lack of discussion on important related works**: Using f-divergence as a regularization in policy optimization has been well studied in many reinforcement learning papers, e.g., Bregman divergence in Mirror Descent Policy Optimization [1], a general surrogate objective for policy optimization [2], to name a few. This paper, however, discusses none of these existing studies. At least the paper should clarify whether/how the theoretical results from the single-agent f-divergence, or even broadly Bregman divergence, can be trivially adapted to the cooperative multiagent RL.

[1] Tomar, M., Shani, L., Efroni, Y. and Ghavamzadeh, M., 2020. Mirror descent policy optimization. arXiv preprint arXiv:2005.09814.
[2] Vaswani, S., Bachem, O., Totaro, S., Müller, R., Garg, S., Geist, M., Machado, M.C., Castro, P.S. and Roux, N.L., 2021. A general class of surrogate functions for stable and efficient reinforcement learning. arXiv preprint arXiv:2108.05828.

### Contribution & quality
**Most of the technical contributions are copied verbatim from or shared with the DPO paper (a separate work as explicitly pointed by this paper, used as a baseline to compare against), which can be flagged as plagiarism**: many of the technical contributions (and even the texts) of this paper are surprisingly the same as the DPO paper, referenced and used for comparison. This will cause ethical issues. See flag for ethical review. (this is rather embarrassing, even though these two papers might come from the same group. As long as they have been claimed as two separate works, they should differ substantially)


**The proposed TVPO is rather similar to the DPO. It is thus hard to believe that the former would perform better than the latter**: the empirical results are actually very weak to show that the TVPO is better than IPPO & DPO. They are not sufficient enough to convince the benefits of the proposed method. Specifically, (a) the performance curves of these three algorithms in Figure 2 are interweaved to each other. There are no noticeable significance among them. Also, why the IPPO plateaus after first million steps in line control. The results in Figure 3 also give the similar performance report for all the three algorithms. (b) The hyper-parameter d_target has been used for both TVPO and DPO, which should play crucial role in the final performance. Fine-tuning d_target might lead to a dramatically different performance in various environments. I’m wondering whether DPO & IPPO is also fine-tuned for a fair comparison.  (c) The motivation of using total variation divergence, instead of KL, is because of its tighter bound. This holds in theory, but may not be necessarily true in practical since both the divergence need to be estimated with integration over the whole action space (continuous). Thus, it is quite important for the paper to show more experiments and details, as mentioned above, to clarify the importance of explicitly optimizing the TV between the last and updated policies.

**The paper holds a bit conflicting assumption for the theoretical guarantee that each agent could receive the central state**. This assumption sounds a bit artificial since, in decentralized learning, ab important factor is the agent’s lack of full information, i.e., the inputs to each agent should be partially observable. The paper should discuss why studying the independent learning with the use of central state information can be important and useful.

**Questions:**

why does this paper copy-pastes a large portion from the DPO paper?

**Details Of Ethics Concerns:**

**This paper copy-pastes verbatim a large portion of the texts from one of its referenced paper**: Kefan Su and Zongqing Lu. Decentralized policy optimization. arXiv preprint arXiv:2211.03032, 2022b. From the description of this paper, "We compare TVPO with four representative fully decentralized learning methods:.... DPO (Su & Lu, 2022b). The empirical results show that TVPO outperforms these baselines in all evaluated tasks,....", this submission is a separate work from DPO. So it is suspicious of plagiarism.

In particular, the **related work on *CDTE* and *Fully Decentralized Learning* is almost the same in these two papers**. On technical part, the ***Fully decentralized critic* is a verbatim copy from the DPO paper**. The ***practical algorithm of TVPO* shares a rather similar idea from the DPO algorithm**.

---

### Official Review · Reviewer_5FWA · 2023-11-04

**Soundness:** 1 poor
**Presentation:** 2 fair
**Contribution:** 2 fair
**Rating:** 3
**Confidence:** 3

**Summary:**

This paper proposes a framework of independent policy optimization in the decentralized multi-agent cooperative RL setting. The algorithm is similar to running PPO on each agent, but the KL divergence is replaced with a general f-divergence. The work first shows the limitation of independent f-divergence policy optimization --- the system may converge to sub-optimal solution even in the two-player case. Then they theoretically show that when the f-divergence is instantiated as TV distance, then together with a large enough regularization coefficient, the payoff will monotonically increase over iterations. In the practical version, they find that the when the regularization coefficient is too large, the step size will be too small and the algorithm will not work. Therefore, in the experiment they use a decreasing regularization. The experiment show that the proposed TVPO algorithm constantly outperform other decentralized algorithms.

**Strengths:**

- The main contribution is the finding that using TV distance in replace of the KL divergence in independent PPO will lead to a better performance in cooperative and decentralized multi-agent RL.

**Weaknesses:**

- Though the paper provides some theory to argue the advantage of using TV distance, I found them not convincing. First, the TV distance is a first-order distance measure, which is non-smooth. When using them as the regularizer in the policy optimization framework, and with a large enough regularization coefficient, it can easily trap the learner's policy, making it not able to update. Imagine performing a mirror ascent with the update rule $\pi_{t+1}=\text{argmax}_\pi  \langle  \pi, Q\rangle - \omega||\pi - \pi_t||_1$ with a large coefficient $ \omega \geq 2||Q|| _\infty$ (this condition is exactly used in Lemma 2). Basically, $\pi _{t+1}$ can only be $\pi _t$. To see this, notice that $ \langle  \pi _{t+1}, Q\rangle - \omega||\pi _{t+1} - \pi_t||_1 \geq  \langle  \pi _{t}, Q\rangle $ implies $\omega || \pi _{t+1} - \pi _t ||_1 \leq \langle \pi _{t+1} - \pi _t, Q \rangle \leq || \pi _{t+1} - \pi _t || _1 || Q || _\infty$, and thus $|| \pi _{t+1} - \pi _t ||_1 > 0$ cannot hold (otherwise $\omega \geq 2||Q|| _\infty$ is violated). I think this is the real reason underlying the convergence result in Lemma 2 and Theorem 1. However, this algorithm is clearly not going to work. That also explains why the authors discover in the experiment that they have to use smaller step size. Thus, I feel that Section 4.2 does not provide the correct reasoning why their algorithm is superior.
- It's nice to see that the proposed TVPO outperforms existing algorithms experimentally. However, since there is no convincing explanation other than the somewhat misleading one mentioned in the previous point, I'm unable to understand where the improvement comes from.

**Questions:**

See the weakness part.

---

### Official Review · Reviewer_s7nG · 2023-11-05

**Soundness:** 3 good
**Presentation:** 3 good
**Contribution:** 3 good
**Rating:** 6
**Confidence:** 3

**Summary:**

This paper proposed a novel independent learning algorithm TVPO for MARL based on f divergence policy optimization. The paper established convergence of TVPO, and provided a practical algorithm for simulation. Extensive simulations are conducted to show the improvement of TVPO compared with other state-of-the-art MARL algorithms.

**Strengths:**

The paper is well-written. The algorithm design intuition is written in a very clear way. I also appreciate the discussion of the practical implementation of TVPO. Simulation results are extensive and demonstrate the improvement of the proposed algorithm compared with the SOTA.

**Weaknesses:**

Q1: what type of convergence results can be obtained for the practical algorithm of TVPO? Section 4.3 is hand-waving and a little confusing, so I hope the authors can provide more comments on this.

Q2: what's the updating rule of beta_i? Can beta_i be different for different i?

**Questions:**

See above.